

# Diatom-oxygen isotopic record from high-altitude Petit Lake (2200 m a.s.l) in the Mediterranean Alps: shedding light on a climatic pulse at 4200 cal. BP

Rosine Cartier[1,2], Florence Sylvestre[1], Christine Paillès[1], Corinne Sonzogni[1], Martine Couapel[1], Anne
Alexandre[1], Jean-Charles Mazur[1], Elodie Brisset[3,4], Cécile Miramont[2], Frédéric Guiter[2]

[1] Aix-Marseille University, CNRS, IRD, Collège de France, INRA. CEREGE, Europôle de l'Arbois, 13545 Aix-en-Provence, France
[2] Aix-Marseille University, CNRS, IRD, Avignon University, IMBE, Europôle de l'Arbois, 13545 Aix-en-Provence, France
[3] IPHES, Institut Català de Paleoecologia Humana i Evolució Social, Tarragona, Spain
[4] Àrea de Prehistòria, Universitat Rovira i Virgili, Tarragona, Spain

*Correspondence to*: Rosine Cartier (cartier.rosine@gmail.com)

**Abstract.** The 4.2 kyrs event, used as a marker of holocene stratigraphy, has been described as a rapid climate change in the northern hemisphere triggering droughts in the Mediterranean region. However, the severity and geographical extent of this event are still the subject of investigation considering the small number of palaeoclimatic records for this time period, and the presence of contrasted climatic expressions between areas. At Petit Lake (France, Mediterranean Alps, 2200 m a.s.l) a multiproxy study of Holocene lake sediments has revealed major changes in erosion processes and phytoplanktonic assemblages in the lake ecosystem around 4200 cal. BP. According to pollen analysis, deforestation is unlikely to be the main explanation of environmental changes as the watershed was covered by open vegetation for the duration of the study period. To test the implication of climate, our study presents an analysis of oxygen isotopes ($\delta^{18}O$) in diatoms describing hydrological modalities during the 4.2 kyrs event in the Mediterranean Alps. The highest values of $\delta^{18}O_{diatom}$ occur from 4400 to 3900 cal. BP and are interpreted as an increase in water evaporation and/or a decrease in freshwater inputs to the lake system. Changes in water balance might have been associated with a change in precipitation sources towards a greater influence of precipitation coming from the Mediterranean area. These results are concomitant to an increase in erosion in the watershed and high representation of very low-dispersal pollen in the sediments suggesting the presence of intense runoff. This new isotopic record together with previously-published proxy-data, allows us to describe the 4.2 kyrs event at Petit Lake as an increase in Mediterranean climate influences in the region, amounting to a general dry period punctuated by episodes of intense runoff occurring on the catchment slopes.

## 1 Introduction

Since the last glaciation, several abrupt climatic changes, each of which had large environmental effects, were identified from palaeoclimatic records (Berger and Guilaine, 2009; Magny et al., 2009). Two of the most important cold events are





recorded in ice cores and in numerous worldwide palaeoenvironmental records. These are: the Younger Dryas (13.500-11.500 cal. BP) at the end of the Late Glacial, and the 8.2 kyrs event in the beginning of the Holocene (Alley et al., 1997; Brauer et al., 1999; Tinner and Lotter, 2001). Other climatic events were described during the Holocene but were interpreted as less intense or regionally limited.

However, although their geographical extent is still under discussions, these climatic events triggered some substantial impacts on the environment. One of them, the "4.2 kyrs event" has been recognised in various studies as an abrupt climate change (Bond et al., 2001; Booth et al., 2005; Huang et al. 2011; Thompson et al., 2002; Staubwasser et al. 2003) which is now commonly used as a marker of Holocene stratigraphy (Walker et al., 2012). In the Mediterranean area, the 4.2 kyrs event is recorded as a complex period lasting a maximum of several hundred years, with contrasted palaeohydrological

expression between regions (Bruneton et al., 2002; Digerfeldt et al., 1997; Drysdale et al., 2006; Kharbouch, 2000; Magny et al., 2009; Miramont et al., 2008; Zanchetta et al., 2011). In the Eastern Mediterranean, this climatic event is recognised to be responsible for severe droughts and was likely involved in the fall of the Akkadian civilisation (Weiss, 1993; Dean et al., 2015; Cullen et al., 2000). In Central Mediterranean, while speleothems from southern Italy (Renella, Corchia Cave) recorded dry conditions from ca. 4300 cal. BP to 3800 cal. BP, dry conditions were less expressed in records from northern

Italy. In the Alps, an opposite trend has been described, and the same time period is characterised by cool and wet conditions (Zanchetta et al., 2011, 2016). Sedimentary records of past lake levels also mirrored changes during this period showing different climatic expression along a latitudinal gradient. At Ledro Lake and Accesa Lake in Italy (respectively 45° N and 42° N) the transition from mid to late Holocene (ca. 4500 cal. BP) is recorded as a transition period towards higher lake levels. However, the opposite trend has been found for the same period at Preola Lake in Sicily (37° N) (Magny et al., 2012).

Finally, a high-resolution record (Accesa Lake, Italy) allowed to interpret the 4.2 kyrs climatic event as a tripartite climatic oscillation characterised by a phase of drier conditions from 4100 to 3950 cal. BP bracketed by two phases of wetter conditions from 4300 to 4100 cal. BP and from 3950 to 3850 cal. BP (Magny et al., 2009). Overall, new palaeoclimatic records from different longitudes and altitudes in the Mediterranean area are needed to better constrain the regional expression of the 4.2 kyrs event.

In the Southern Alps, the high-altitude Petit Lake (Massif du Mercantour, France, 2200 m a.s.l) offers pollen and diatom-rich sediments covering the last 5000 years. A multiproxy analysis, including sedimentological and geochemical measurements (XRF, ICP-AES) as well as pollen and diatom morphological analysis clearly revealed two phases separated by a major shift around 4200 cal. BP. This major shift was characterised by a detrital pulse (Brisset et al., 2012, 2013) followed by a long-lasting change in tychoplanktonic diatom assemblages (Cartier et al., 2015). Over the last 5000 years, the vegetation around

the lake has been reconstructed as open, rejecting the hypothesis of a massive deforestation in the catchment as the explanation for the detrital pulse. Therefore, the involvement of a rapid climate change either in precipitation regime or temperature leading to increasing soil erosion and runoff around 4200 cal. BP was proposed (Brisset et al., 2012, 2013; Cartier et al., 2015).





In order to test the latter hypothesis, we measured the oxygen isotopic composition ($\delta^{18}O$) in diatoms ($\delta^{18}O_{diatom}$) taken from the Petit Lake 5000-present day sedimentary core previously used for multiproxy reconstruction. Because $\delta^{18}O$ values are a function of lake water isotopic composition ($\delta^{18}O_{lake}$) and temperature, $\delta^{18}O_{diatom}$ records are commonly used for climatic reconstructions (e.g. Barker et al., 2001; Leng et al., 2006; Quesada et al., 2015). Previous $\delta^{18}O$ records from Mediterranean

lakes were discussed in terms of changes in precipitation and lake water balance, which depend on lake location and watershed properties (Roberts et al., 2010). Here, from the $\delta^{18}O_{diatom}$ record we aim to assess the last 5000 years of hydrological changes at Petit Lake and build assumptions on climatic changes that may have occurred around 4200 cal. BP in the Southern Alps.

## 2 Site settings

In the Mercantour range, alpine and mediterranean influences produce a climate marked by mild winters and dry summers. Mean annual temperature at 1800 m a.s.l. is 5 °C, varying from 0.3 °C in winter to 9.9 °C in summer (Durand et al., 2009), with rainfall occurring mainly in spring and autumn. Mean annual precipitation is 1340 mm at 1800 m a.s.l. Snow depths in winter are relatively important (150 to 250 cm at 2400 m a.s.l.) due to moisture from the nearby Mediterranean Sea. Snow cover duration is about 185 days at 2100 m a.s.l. mainly from November to April (Durand et al., 2009).

Petit Lake (2200 m a.s.l; N 44°06.789; E 7°11.342) is a small circular body of water 150 m in diameter located in the Southern French Alps about 60 km from the Mediterranean Sea. Petit Lake is at the lowest elevation of a chain of five lakes that were partly formed by glacier retreat (Fig. 1). The lakes are connected in the spring by ice meltwater but remain unconnected for the rest of the year. The lake catchment (area: 6 km$^2$) culminating at 2600 m a.s.l. is composed of crystalline bedrock (gneiss and migmatites) and is largely covered by alpine meadows; the upper tree line (*Larix* sp.) being located at

about 2100 m a.s.l. The lake surface is usually frozen from October to April. The depth of Petit Lake is up to 7 m in the wake of the snow-melt in late spring and decreases to 6.5-6 m at the end of summer.

Because it is located at the extreme south-west of the Alps, Petit Lake is strongly influenced by precipitation originating from the Mediterranean region during the summer, and by precipitation from the Atlantic in the winter (Bolle, 2003). In Southern France, precipitation is mostly generated by the clash between the warm, humid air of Mediterranean or mixed

Atlantic-Mediterranean origin and cool air masses coming from the North (Celle-Jeanton, 2000). Precipitation of Mediterranean origin has a weighted annual mean ($\delta^{18}Op$) of −4.33 ‰ (standard deviation s=1.72 ‰), and precipitation from the Atlantic has a $\delta^{18}Op$ of −8.48 ‰ (standard deviation s=3.51 ‰) (from April 1997 to March 1999; Celle-Jeanton et al., 2004). This is reflected in the seasonal weighted mean $\delta^{18}Op$ in the Alps (Fig. 1), to which is added an altitude effect of -0.2 ‰ per 100 m (Ambach et al., 1968). Figure 2 shows monthly weighted means of $\delta^{18}Op$ from GNIP stations around Petit

Lake (IAEA/WMO, 2018; Thonon-les-Bains: N 46°22, E 6°28; Draix: N 44°13, E 6°33; Malaussène: N 43°92, E 7°13; Monaco: N 43°73, E 7°42). For Thonon-les-Bains (385 m a.s.l) and Draix (851 m a.s.l), two stations north-east to Petit Lake, mean $\delta^{18}Op$ during summer months is -7.4 ‰ and -11.3 ‰ during winter months. South of Petit Lake and closer to the



Mediterranean Sea, the mean $\delta^{18}O_p$ at Malaussène station (359 m a.s.l) is -5.8 ‰ during summer months and -4.9 ‰ during winter months; and -2.18 ‰ and -5.85 ‰ for Monaco (2 m a.s.l) (Fig. 2). At these stations, $\delta^{18}O$ values are not a function of the amount of precipitation but rather varies according to the season. $\delta^{18}O_p$ is low during periods of cooler air temperatures according to a linear relationship (IAEA/WMO, 2018). Two lake water samples were collected for isotopic measurements at

two times during the year: once in spring (May 17th, 2011) after the snow had melted, and once at the end of the summer (September 17th, 2011). The respective $\delta^{18}O$ and $\delta D$ compositions are -11.35 ‰ and -80.36 ‰; and -10.19 ‰ and -72.6 ‰ (Fig. 2).

## 3 Material and methods

Sediment core PET09P2 (144 cm-long) was sampled in 2009 in the deepest part of the lake using a UWITEC gravity corer.

Core PET09P2 is organic-rich (total organic carbon represents 9 % of the dry weight on average) and has a high abundance in biogenic silica (averaging 65 % of the dry weight) (Brisset et al., 2013). Diatoms (D) represent the major contribution of biogenic silica in the sedimentary record. Only a few cysts of Chrysophyceae (C) were identified (C/D ratio = 0.01). The age-depth model covering the last 4800 years is based on short-lived $^{210}$Pb and $^{137}$Cs radionuclides and seven $^{14}$C ages obtained from terrestrial macro-remains (see Brisset et al. [2013] for further details).

Twenty diatom samples (1 cm$^3$) were sub-sampled from core PET09P2. Each diatom sample of 1cm$^3$ includes on average 36 years (min: 11 years; max: 55 years) of sedimentation according to the age-depth model. Diatom samples were weighed after drying at 50 °C. To remove carbonates and organic matter, the samples were first treated using standard procedures (bathed in a 1:1 mixture of $H_2O_2$: water, a 1:1 mixture of HCl: water, and repeatedly rinsed in distilled water). Following these steps, the identification and counting of diatom species for palaeoenvironmental reconstruction were performed; and the data were

reported in Cartier et al. (2015). Then, diatom silica was cleaned from remaining detrital particles by following the protocol developed by Crespin et al. (2008), which includes 7 steps based on chemical treatments and physical separation. This protocol has already been used successfully in previous studies (Alexandre et al., 2012; Crespin et al., 2010; Quesada et al., 2015). The purity of each sample was then checked using optical and scanning electron microscopy (SEM) together with micro-X-ray fluorescence (XRF) measurements (5 measurements per sample). The hardware used for these analyses

consisted of a HORIBA XGT-5000177 microscope equipped with an X-ray guide tube capable of producing a focused, high-intensity beam having a 100 μm spot size (detection limit: 2 ppm). The following compounds were detected via XRF: $SiO_2$, $Al_2O_3$, $K_2O$, $CaO$, $TiO_2$, $Fe_2O_3$, and $Br_2O$. The samples are on average consisted of 97.2 % (s=1.8 %) of $SiO_2$. SEM observations showed no visible remains of detrital particles or organic matter (see picture Fig. 3C). The diatoms themselves were very well preserved and showed only minor signs of dissolution.

Measurements of oxygen isotopes from diatoms were performed at CEREGE Stable Isotope laboratory (Aix-en-Provence, France) by performing the following sequence of steps. Firstly, the samples were placed an inert Gas Flow Dehydration (iGFD) apparatus, adapted from Chapligin et al. (2010), and were dehydrated by ramp degassing (2 h heating to 1020 °C, 1.5



h held constant at 1020 °C, 2h cooling down to 400 °C) under a continuous dry $N_2$ flow. Oxygen extractions were then performed using the IR Laser-Heating Fluorination Technique (Alexandre et al., 2006; Crespin et al., 2008). No ejection occurred during the analysis. The oxygen gas samples were sent directly to and analysed by a dual-inlet mass spectrometer (ThermoQuest Finnigan Delta Plus). Measured $\delta^{18}O$ values were corrected on a daily basis using a quartz lab standard

($\delta^{18}O_{Boulangé\ 50-100\ \mu m}$) calibrated on NBS28 (9.6 ± 0.3 ‰; n=11) (Alexandre et al., 2006; Crespin et al., 2008). The values were expressed in the standard δ-notation relative to V-SMOW. The long-term precision of the quartz lab standard is ± 0.2 ‰ (1s; $n$=50). The final $\delta^{18}O$ values for each sample is the average of two replicates, yielding a reproducibility of better than ± 0.2 ‰ (1σ). The age-depth model of PET09P2 (Brisset et al., 2013) was constructed using the R package Clam (Blaauw et al., 2011). For the purpose of this study, we have recalculated the age-depth model using the Bacon R package (Blaauw and

Christen, 2011) and implemented the function "proxy.ghost" (square resolution:200) in order to take into account the chronological uncertainties associated with the proxy representation. The result is a range of possible ages for each sample depth, each of which is represented on the graph. The darkest grey is assigned to the most likely value within the entire core (normalised to 1). Lower age probabilities are coloured in lighter grey.

### 4 Results

Oxygen isotopes values (‰ vs V-SMOW) measured on the 20 sedimentary diatom samples (table 1) are plotted against ages (cal. BP) and presented in Fig. 3A. $\delta^{18}O_{diatom}$ values ranged from 26.6 to 32 ‰ with an average value of 30 ‰. The standard deviation for each measurement is shown by error bars on Fig. 3A.

From 4800 to 4400 cal. BP, the $\delta^{18}O_{diatom}$ average value is 30.3 ‰. The lowest value during this period is for the sample at 4750 cal. BP with a $\delta^{18}O_{diatom}$ value of 28.97 ‰. At 4400 cal. BP, $\delta^{18}O_{diatom}$ increases quickly and reached a maximum value

of 31 ‰. $\delta^{18}O_{diatom}$ remains high between 4400 and 3900 cal. BP, and decreases to values below those observed at the base of the core afterwards. From 3900 to 700 cal. BP, $\delta^{18}O_{diatom}$ shows low amplitude variations with an average value of 29.6 ‰. Three samples have higher values at 2600, 1600 and 1100 cal. BP. After 700 cal. BP, the $\delta^{18}O_{diatom}$ falls sharply to its lowest value over the study period (26.6 ‰ at 309 cal. BP). The latest value at 2.5 cm depth (1986 AD) increases again but remains low (27.8 ‰) compared to previous periods (Fig. 3A).

A zoom on the period 4800-3000 cal. BP period, taking into account the uncertainties of the age-depth model is presented in Fig. 3B. Four $^{14}C$ ages (Fig. 3A) exist in this time interval, yielding an age-depth model precision of 200 years. The highest values of $\delta^{18}O_{diatom}$ occur between 4500 and 3800 cal. BP. The most likely ages for this period are from 4400 to 3900 ca. BP, i.e. corresponding to a 500-year period.





## 5 Discussion

### 5.1 Climatic interpretation of the $\delta^{18}O_{diatom}$ record

Effects of human occupation on ecosystems of the Mediterranean Alps, since the middle Holocene, have been widely documented (De Beaulieu, 1977; Mocci et al., 2008; Walsh et al., 2007; Walsh and Mocci, 2016). Human impacts can
contribute to changes in erosion and vegetation and along with climate change, must be taken into account when interpreting erosion and vegetation proxies (Giguet-covex et al., 2011; Jalut, 2009; Roberts et al., 2010; 2011). By contrast, lake temperature and hydrological balance as reflected by $\delta^{18}O_{lake\ water}$ are the only factors governing changes in $\delta^{18}O_{diatom}$. $\delta^{18}O_{lake\ water}$ itself depends on the $\delta^{18}O$ composition of the moisture source, the precipitation amount and regime, and in a lesser extent, the transport pathway followed by moving air masses (Dansgaard, 1964; Gat, 1996). These factors, that are inherent
to the climate system, can be evaluated from an accurate interpretation of the high $\delta^{18}O_{diatom}$ values from 4400 to 3900 cal. BP at Petit Lake. In addition, the lake hydrological balance has to be assessed. At a minimum, potential isotope fractionation related to the dissolution of the diatom spicules during sedimentation may occur (Dodd et al., 2017). However, SEM observations of our diatom samples did not show significant dissolution features (Fig. 3C).

Water inflows to Petit Lake consist of direct precipitation (rain and snow) and intermittent streams that form during the
spring ice melt. The outlet of Petit Lake (today blocked by a dam) is an intermittent surface outlet and is non-active when the lake level is decreasing of 1 meter (Figure 1). Therefore, the hydrological regime alternates between two states: an open system when the outlet is active during snow melt and a closed system during summer months when most water losses are due to evaporation. In these systems, changes in water balance are commonly recognised to be the main factor triggering large excursions in the isotopic signal (Leng and Barker, 2006; Roberts et al., 2008). Summer months correspond to the
season during which diatoms grow after the first spring blooms. During this time of year and given the shallowness of the lake, waters are well mixed under the effect of wind (Cartier, 2016). From that perspective, it is reasonable to assume that an increase in lake water evaporation, due to drier climate conditions during the summer months, may have triggered an increase in $\delta^{18}O_{diatom}$. A rapid change in water conditions around 4200 cal. BP is supported by the dominance of the diatom species *Staurosirella pinnata* (Fig. 4), a species with high tolerance to rapid changes in alkalinity and conductivity in
unstable aquatic environments (Cartier et al., 2015).

An increasing contribution of precipitation coming from the Mediterranean area, with high $\delta^{18}Op$, can also contribute to an increase in $\delta^{18}O_{lake\ water}$ and consequently in $\delta^{18}O_{diatom}$. Today, Mediterranean precipitation favours runoff and erosion in steep areas (Kosmas et al., 2002). Geochemical data showing high terrigenous inputs to Petit Lake between 4400 and 4000 cal. BP (Fig. 4), interpreted as an increase of runoff in the watershed (Brisset et al., 2013), are thus consistent with a greater seasonal
variability of the Mediterranean climate characterised by intense precipitation occurring in fall and spring and significantly drier periods in the summer months (Durand et al., 2009).

A decrease in lake water temperature could be suggested, additionally, to explain an increase in $\delta^{18}O_{diatom}$ fractionation. A 2 ‰ increase in $\delta^{18}O_{diatom}$ (variation of 5 ‰ for the entire record; fig. 3A) would imply a lake water temperature drop of 10 °C



according to the thermo-dependent fractionation coefficient between temperature and diatoms of -0.2 ‰/°C (Brandriss et al., 1998; Crespin et al., 2010; Moschen et al., 2005). This inferred-temperature is not in agreement with air temperature estimates based on chironomids and pollen assemblages from the Swiss Alps and Europe which suggest that temperature variations did not exceed 2 °C during the Holocene (Davis et al., 2003; Heiri et al., 2003). Thus, the $\delta^{18}O_{diatom}$ shift cannot be

explained by a drop in summer month temperatures and temperature doesn't appear to be the main factor of $\delta^{18}O$ variability at Petit Lake. In summary, the rapid increase in $\delta^{18}O$ diatom from 4400 to 3900 cal. BP is most likely the result of an increase in water evaporation possibly associated with a shift in precipitation origin and distribution over the year. This state lasted for ca. 500 years.

After 3900 cal. BP, $\delta^{18}O_{diatom}$ values decreased and remain relatively constant for 3300 years suggesting less water

evaporation/humid conditions during the Neoglacial period. However, the low resolution of the record in this part might limits the identification of short-term events. A last major excursion impacting the $\delta^{18}O_{diatom}$ record, but not the other proxies, is recorded around 310 cal. BP. It consists of a rapid drop in the $\delta^{18}O_{diatom}$ values (Fig. 3A). Conversely to what may have happened during the time interval 4400-3900 cal. BP, lower evaporation during the summer months, increasing precipitation from Atlantic sources, and/or an increase in summer air temperature, appear plausible. This time span falls within the Little

Ice Age (450-50 cal. BP), which is known to be a cold and humid period in the Southern Alps according to tree-ring records (Corona et al., 2010), fluvial activity reconstruction (Miramont et al., 1998; Sivan et al., 2006) and glacial tongue advances (Holzhauser et al., 2005; Ivy-ochs et al., 2009). Therefore, the decrease in $\delta^{18}O_{diatom}$ at Petit Lake is most likely the response of increased humidity and Atlantic precipitation influences. These results are concomitant with a strong decrease in $\delta^{18}O$ measured on ostracods from Allos Lake sediments (Cartier et al., in prep) indicating the expression of a regional climate

change. Other cold periods such as the Late Antiquity recorded ca. 1700 cal. BP in Southern France were not identified in the record, suggesting that the climatic effects of the LIA were of a greater magnitude in the studied area. However, better resolution is needed to confirm this observation.

## 5.2 Expression of the 4.2 kyrs climatic event and regional comparison

At Petit Lake the highest values of $\delta^{18}O_{diatom}$, from 4400 to 3900 cal. BP were broadly interpreted as an increase in water

evaporation/decrease in water inputs to the lake. A strong influence of Mediterranean precipitation characterised by higher $\delta^{18}Op$ might also have been an additional factor favouring increased $\delta^{18}O_{lake}$ at this time. By considering the other palaeoenvironmental proxies, these results appear concomitant to high terrigenous inputs to the lake and chemical weathering of soils (Fig. 4). These minerogenic inputs together with a high representation of very low-dispersal alpine meadow pollen were interpreted as the result of intense runoff on the catchment slopes (Brisset et al., 2013, Fig. 4). In the

lake ecosystem, the dominant diatom species, *Staurosirella pinnata*, was then replaced by other species like *Pseudostaurosira robusta* (Fig. 4). Results provided by measuring the oxygen isotopic composition of fossil diatoms strongly support the hypothesis of a rapid climate change which triggered the shift in the environmental history of the watershed. Local responses to this climatic event probably happened in several stages: (1) a presence of more intense runoff



in a general drier context; (2) increased erosion in the watershed and terrigenous inputs to the lake; (3) a change in lake ecosystem properties (e.g. water transparency, conductivity) and aquatic assemblages. An indirect effect of climate on the lake ecosystem is supported by the presence of an offset between changes in $\delta^{18}O_{diatom}$ and diatom assemblages. Indeed, the shift in diatom species occurred after the first increase in $\delta^{18}O_{diatom}$ and before the decrease of $\delta^{18}O_{diatom}$ showing the end of

the 4.2 kyrs event period. Indirect effects have been more likely modulated by changes in watershed properties like changes in erosion processes as shown in previous studies (Jeppesen et al., 1997; McQueen et al., 1989). To sum up, the 4.2 kyrs climatic event is expressed at Petit Lake as a period of general dry climatic conditions during which intense rainfall occurred. In a Mediterranean context, strong erosion on dry soils is increased by high seasonal variability and the presence of extreme episodes (droughts, flashflood events) particularly conducive to denudation processes (Brisset et al., 2017; Nearing et al.,

2004; Yaalon, 1997). At Petit Lake, the 4.2 kyrs event by changes precipitation regime and water balance, led to a long-lasting change in the lake trajectory.

Close to Petit Lake, a palaeoenvironmental record at Grenouilles Lake also recorded high percentages of Poaceae, Chenopodiaceae and Caryophyllaceae around 4200 cal. BP even if the 4.2 kyrs climatic event was not discussed in the interpretation (Kharbouch, 2000). At Allos Lake, located in the Mercantour Massif and at a similar altitude to Petit Lake,

there is no evidence of a major detrital supply at 4200 cal. BP. However, this period is generally characterised by high lacustrine production (represented by an increase in organic sedimentation rate) and rapid shifts in the percentages of the benthic diatom *Ellerbeckia arenaria* suggesting a high variability in lake levels with potential periods of droughts during the presence of these diatom-rich laminae (Cartier et al., 2018). At Saint Léger Lake (Alpes-de-Haute-Provence), a reconstruction of past lake level changes argues, this time, for a moderate rise of the lake level from 4500 to 3000 cal. BP

(Digerfeldt et al., 1997). Other detrital or flood events have been recorded between 4500 and 3000 cal. BP in the Alps for example at Bourget Lake (Arnaud et al., 2005; 2012) and a review of dated-landslides revealed a cluster around 4200 cal. BP supporting the presence of heavy precipitation (Zerathe et al., 2014; Fig. 5). In the Massif du Mont Blanc (46° N) a moraine contemporary with the 4.2 kyrs climatic event has been dated suggesting a glacier advance during this period (Le Roy et al., 2017). At a broader scale, stable isotope and trace element data from a calcite flowstone located in northern Italy (Buca della

Renella; 44° N) have shown that the 4.2 kyrs event was expressed in mid-latitude Europe by dry climatic conditions (Drysdale et al., 2006) (Fig. 5). A similar trend has been found at Preola Lake (37° N) in Sicily (Magny et al., 2012). However, this period of rapid climatic changes is also documented in some lacustrine records by shifts from low stands to higher lake levels around 4000 cal. BP, for example at Cerin Lake, Ledro Lake (45° N) and Accesa Lake (42° N) (Magny et al., 2013) (Fig. 5).

Overall, few isotopic records with sufficient resolution for studying the 4.2 kyrs climatic event exist in the Western Mediterranean. Most of the records on stalagmites or lake sediments are located in central Italy or in the Eastern part of the Mediterranean region. Concerning the use of other palaeoenvironmental proxies, the difficulty lies in separating local changes in land use from climatic effects. The potential of Petit Lake to record past climatic events is probably enhanced by its location in the head of the watershed and the sparse vegetation which exposes soils to erosion. In addition, the semi-



closed lacustrine system might have increased the responsiveness of Petit Lake to changes in water regime. Local watershed properties, a lack of isotopic records and age-depth model accuracy might explain the difficulties in identifying similar trends during the 4.2 kyrs event, which still requires further attention at a regional scale.

## 6 Conclusion

Measurements of oxygen isotopes in diatoms ($\delta^{18}O_{diatom}$) from Petit Lake were performed on the last 5000 years of the sedimentary record in order to investigate the influence of changes in climate on this alpine watershed. A major and rapid shift in the environmental history was recorded at 4200 cal. BP in both terrestrial and lacustrine proxy-data. The system turned from a steady-state without soil erosion to a new state dominated by a degradation trend for both slopes and vegetation cover. Additionally, a long-lasting change in diatom assemblages highlighted major variations in lacustrine living

conditions. The new $\delta^{18}O_{diatom}$ record for Petit Lake was used to reconstruct past hydrological changes and decipher climatic implications from local human impacts around 4200 cal. BP. Over the study period, $\delta^{18}O_{diatom}$ varied between 26.6 ‰ and 32 ‰ vs V-SMOW with an average of 30 ‰. The highest values of $\delta^{18}O_{diatom}$ (> 31 ‰) stand out from 4400 to 3900 cal. BP, making it possible to identify the climatic expression of the 4.2 kyrs event in the Southern Alps. Then, from 3900 cal. BP to present-day, $\delta^{18}O_{diatom}$ decreased and showed low amplitude variations (mean at 29.6 ‰) except for a major excursion during

the Little Ice Age (309 cal. BP) towards a decrease of the $\delta^{18}O_{diatom}$ to 26.6 ‰.

The highest $\delta^{18}O_{diatom}$ values from 4400 to 3900 cal. BP have been interpreted as the presence of high water evaporation during summer months possibly associated with and a stronger contribution of precipitation coming from the Mediterranean area. Linked to previous palaeoenvironmental studies, these results allow us to describe the 4.2 kyrs event in the Southern Alps as a period of general drier climatic conditions during which intense rainfall occurred on catchment slopes. A higher

seasonal variability of the Mediterranean climate might have triggered the increase in erosion, particularly of the soils that have only a sparse vegetation cover and, secondly, appear to have caused the change in phytoplanktonic assemblages in the lake. In a context where landscapes are already modified by human activities, all necessary conditions for increasing the effects of a rapid climate change (i.e. change in precipitation regime) were present on this alpine watershed. This isotopic record at Petit Lake has revealed the implication of the 4.2 kyrs event in abrupt ecosystem changes in the Southern Alps and

is useful to better understand the intensity and geographical extent of this climatic event in the Mediterranean region.

## 7 Author contribution

Rosine Cartier wrote the manuscript and performed analysis with Florence Sylvestre. Christine Paillès, Frédéric Guiter and Cécile Miramont provided funding support and material. Anne Alexandre, Elodie Brisset, Frédéric Guiter helped improving the manuscript. Corinne Sonzogni, Martine Couapel and Jean-Charles Mazur worked in analysing samples. All the co-

authors gave their comments and agreement during the writing process.




## 8 Competing interests

The authors declare that they have no conflict of interest.

## 9 Acknowledgements

This work was supported by the ECCOREV research federation (HOMERE program led by F. Guiter and C. Paillès). The
PhD thesis work of R. Cartier (Aix-Marseille University) was funded by the French Ministry of Education.
We thank C. Vallet-Coulomb (CEREGE, France) for the oxygen isotope analysis of modern Petit Lake waters and P.
Chaurand (CEREGE, France) for providing help with the micro-XRF measurements. Thanks to A. Tonetto (Aix-Marseille
University) for managing the SEM in Marseille. Coring of Petit Lake (in 2009 and 2012) was made possible thanks to F.
Arnaud (EDYTEM), C. Giguet-Covex (EDYTEM), E. Malet (EDYTEM), J. Pansu (Princeton University), J. Poulenard
(EDYTEM) and B. Wilhelm (LTHE).

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





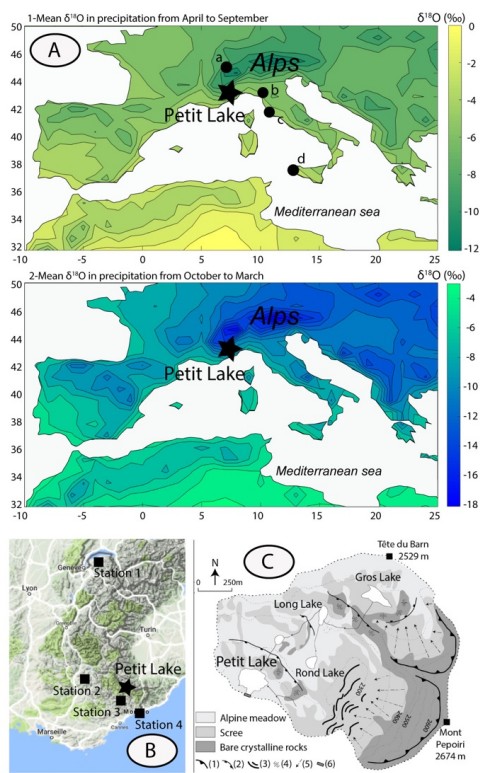

**Figure 1: localisation map of Petit Lake: A) mean δ^18O in precipitation (δ^18Op) (in ‰ vs VSMOW) in the western Mediterranean region (IAEA/WMO, 2018) and selected palaeoclimatic studies (a: Mont Blanc Massif (Le Roy et al., 2017), b: Buca della Renella (Drysdale et al., 2006), c: Accesa Lake (Magny et al., 2009), d: Preola Lake (Magny et al., 2012); B) GNIP stations (IAEA/WMO, 2018) in black squares: 1) Thonon-les-bains, 2) Draix, 3) Malaussène, 4) Monaco; C) watershed characteristics: 1) glacial cirque,2) glacial step, 3) moraine, 4) polished bedrock, 5) active debris slope, 6) dam built in 1947.**





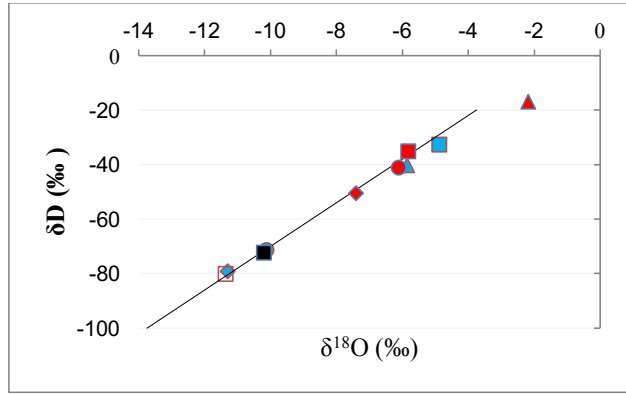

**Figure 2: δ¹⁸O$p$ (in ‰ vs VSMOW) from GNIP stations (IAEA/WMO, 2018) and from Petit Lake at two key times of the year (▯-**
5  **May 17th 2011, ■- September 17th 2011) plotted across the global meteoric water line (black line). Locations of GNIP stations are shown in Figure 1. The mean weighted average of δ¹⁸O$p$ for each station is represented by red dots for summer months (April to September) and blue dots for winter months (October to March). Thonon-les-bains (◆), Malaussène ( ), Monaco (▲), Draix (●).**

25



**Table 1: oxygen isotopes measurements in diatoms (in ‰ vs V-SMOW) for the core PET09P2**

| Sample | Depth (cm) | Age (cal. BP) | $\delta^{18}O_{diatom}$ | St. dev. |
|--------|-----------|---------------|------------------------|----------|
| PET2.5 | 2.5 | -36 | 27.85 | 0.58 |
| PET13 | 13 | 309 | 26.55 | 0.10 |
| PET 21.5 | 21.5 | 744 | 29.31 | 0.07 |
| PET29 | 29 | 1118 | 30.06 | 0.11 |
| PET37 | 37 | 1436 | 29.13 | 0.19 |
| PET45 | 45 | 1666 | 29.74 | 0.12 |
| PET55 | 55 | 1930 | 29.23 | 0.05 |
| PET68 | 68 | 2464 | 30.17 | 0.24 |
| PET78 | 78 | 2996 | 29.07 | 0.35 |
| PET85 | 85 | 3372 | 29.96 | 0.02 |
| PET94 | 94 | 3798 | 29.86 | 0.05 |
| PET100 | 100 | 4018 | 31.34 | 0.35 |
| PET108 | 108 | 4241 | 31.03 | 0.24 |
| PET109.5 | 109.5 | 4275 | 31.97 | 0.23 |
| PET115 | 115 | 4386 | 30.73 | 0.03 |
| PET120 | 120 | 4471 | 30.35 | 0.52 |
| PET127 | 127 | 4570 | 30.36 | 0.05 |
| PET135 | 135 | 4667 | 30.48 | 0.05 |
| PET142 | 142 | 4747 | 28.97 | 0.05 |
| PET144 | 144 | 4770 | 30.73 | 0.12 |



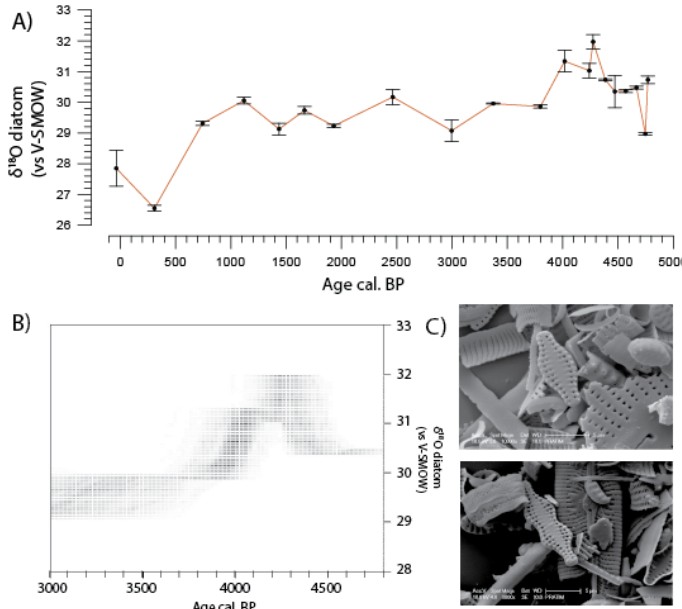

**Figure 3: A) Oxygen isotope composition of diatoms (δ¹⁸Odiatom expressed in ‰ vs V-SMOW) from Petit Lake sediments; B) δ¹⁸O diatom (vs-VSMOW) taking into account the age uncertainties (the darkest grey is assigned to the most likely value within the entire core); C) SEM image of a cleaned diatom sample from 127 cm depth using a Scanning Electron Microscope.**




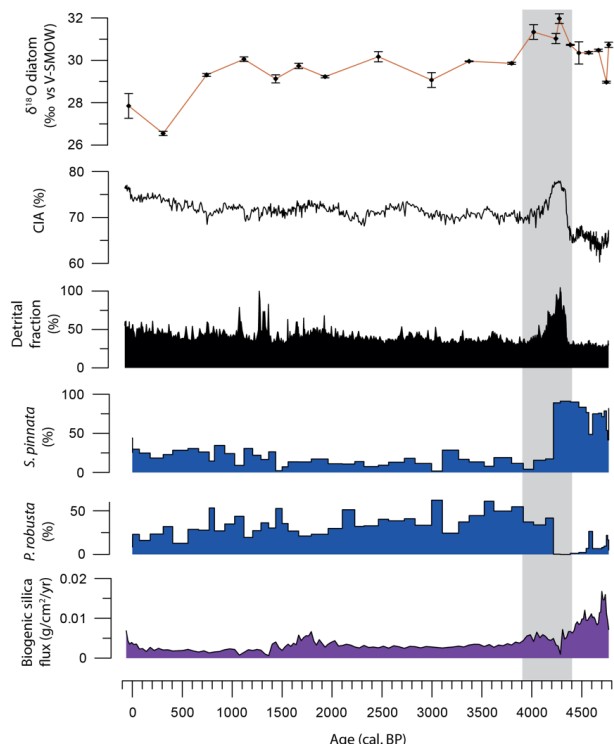

**Figure 4: Multiproxy comparison of environmental responses to the 4.2 climatic event including oxygen isotopes measurements on diatoms (δ18O diatom, ‰ vs V-SMOW, this paper), the Chemical Index of soil Alteration (CIA; Brisset et al., 2013), the detrital fraction (% dry weight), biogenic silica flux (g.cm$^2$.yr; Cartier et al., 2015), and dominant diatom species (relative abundance (%) of *S. Pinnata*, *P. robusta*).**





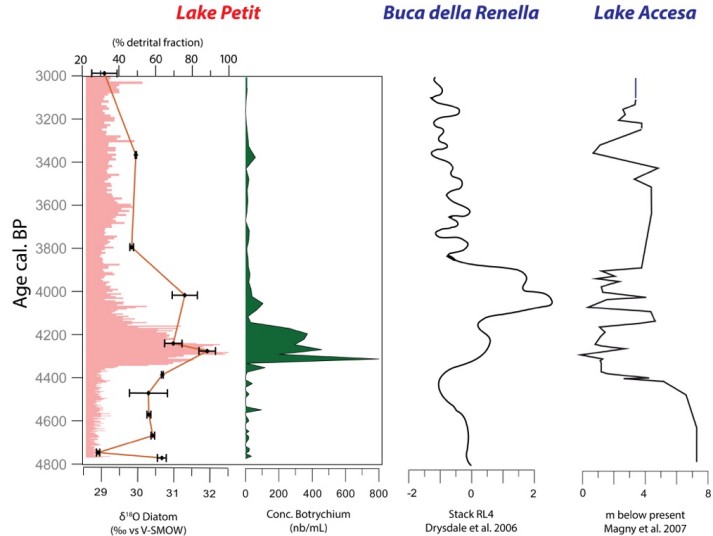

**Figure 5: Oxygen isotopes measurements in diatoms ($\delta^{18}O_{diatoms}$ ‰ vs V-SMOW; this work), detrital fraction (%) and conc.**
***Botrychium* (nb/mL) (Brisset et al., 2015) at Lake Petit compared to the palaeoclimatic record at Buca della Renella (northern**
5  **Italy, Drysdale et al., 2006) and Lake level at Accesa (central Italy, Magny et al. 2007).**