# Peer review of "Diatom-oxygen isotopic record from high-altitude Petit Lake (2200 m a.s.l) in the Mediterranean Alps: shedding light on a climatic pulse at 4200 cal. BP"

_Climate of the Past, 2018_

## Referee Comment (RC1) · Anonymous Referee #1 · 27 Sep 2018

This paper presents new data from ∼4.2ka from a part of the world where data from this time are lacking. The data collection (e.g. purification of diatom samples) and the treatment of uncertainty in the age model is thorough and allows us to have more confidence in the results. The introduction presents a good hypothesis based on the previously produced proxies to test in this paper with the d18Odiatom data.

However, my main concern is regarding the interpretation of the (slight) increase in d18Odiatom around 4.2ka as primarily a water balance signal (i.e. indicating a shift to drier conditions). If the isotope shift were of a higher magnitude, then it would be possible to have more certainty in the interpretation. But, and the authors acknowledge
this at points, a change in precipitation source could also account for some of the isotope shift. In fact, potentially it could account for all of the isotope shift. Also, a decrease in snow (with its very low d18O) around 4.2ka with everything else staying the same could account for the d18Odiatom rise. While your argument about temperature not being the main driver if valid, more thought and caution needs to go in to your interpretation. There are only a couple of modern day lake water isotope values, but even the summer one is fairly low, so how evaporatively-driven is the isotope system? While I agree that something definitely happened in the lake ∼4.2ka as the d18Odiatom changes are outside of uncertainty and other proxies show changes too, with so few d18Odiatom data points and the relatively small magnitude of the change means it is difficult to unequivocally say that a change to drier conditions, rather than a change in precipitation source, or decrease in snow, or a combination of these factors, was responsible for the d18Odiatom change.

Therefore, I think the argument of the driver(s) of d18Odiatom needs to be more cautious and more thought through.

Nevertheless, this is a valuable new dataset that is robustly analysed and adds to our knowledge of what was going on in the Mediterranean region around 4.2ka, so I support its publication if my points are addressed.

Specific comments: First line of abstract: Holocene not holocene Page 2 line 5: discussion not discussions Page 2 line 13: change to "In the Central Mediterranean..." Good summary of previous literature. In addition to Walker et al. 2012, could now reference the new Holocene subdivisions brought in this year. Detailed purification strategy and XRF check to ensure contamination removed. Good SEM work to check diatoms not dissolved. Good, thorough treatment of uncertainty with age model. Page 6 line 7 "...temperature and d18Olakewater..." rather than "temperature and hydrological balance as reflected by d18Olakewater..." Page 7 line 5 "does not" not "doesn't" Good discussion of difficulties of comparing local and climate signals in other records from the region from 4.2ka time. Diatom samples look clean as no correlation between

d18Odiatom and %contamination.

In answer to the review criteria questions: 1. Yes 2. Yes, new data 3. Yes, although need to be more cautious about whether evaporation is the major driver of d18Odiatom. 4. Yes 5. Yes 6. Yes 7. Yes 8. Yes, although I'm not sure of the term 'climatic pulse'. Something like 'event' or 'interval' might be better. 9. Yes 10. Yes 11. Mostly 12. Yes 13. Yes – see my comments on d18Odiatom interpretation 14. Yes 15. Yes

---

## Referee Comment (RC2) · Anonymous Referee #2 · 10 Oct 2018

Review for: Diatom-oxygen isotopic record from high-altitude Petit Lake (2200 m a.s.l) in the Mediterranean Alps: shedding light on a climatic pulse at 4200 cal. BP

Authors: Rosine Cartier, Florence Sylvestre, Christine Paillès, Corinne Sonzogni, Martine Couapel, Anne Alexandre, Jean-Charles Mazur, Elodie Brisset, Cécile Miramont, Frédéric Guiter

General comments: Cartier et al. present a novel diatom $\partial$18O dataset spanning the past ∼5000 yrs from Lake Petit in the SW French Alps. The focus of the study lies in the local to regional characterization of hydroclimate perturbations around the 4.2ka climatic event and is thus relevant within the scope of CP. The oxygen isotope data

(4 data points for the time slice) show a clear excursion towards higher values, which the authors primarily interpret to be the result of drier conditions with increased evaporation in the Lake Petit watershed. By using already published data from a previous 'multiproxy' study of the Petit sediment record the authors further suggest that the period was characterized by precipitation induced flood events. While the interpretation of the oxygen isotope data appears mostly sound (detailed comments below) it is sometimes hard to follow the argumentation regarding the sedimentary indicators that suggest a higher frequency of flood events/catchment erosion during this period. Since this is quite a central statement for the hydrological reconstructions I would suggest the authors provide a complete lithostratigraphic account of the record (eg. Are there any discernible or identifiable flood layers?). In a broader sense the manuscript contributes an additional hydroclimatic dataset that will help to paint a regional picture of climate repercussions during the 4.2ka event in the Mediterranean borderlands. The manuscript is in most parts appropriately structured, in some parts appropriately illustrated, but suffers from a large number of spelling mistakes and grammatical flaws.

Specific comments:

Site settings - the seasonal distribution is quite important in this setting. If possible provide precipitation data for summer and winter months. Also, what controls winter snow depth in this setting? From the data presented it seems as if snow depth (by the end of the season?) varies largely from year to year.

- p.4, l. 1-4. Temperature dependent fractionation of rainfall is suggested as the main driver of seasonal oxygen isotopic composition. However, $\partial18O$ of precipitation at Malaussene is lower (by almost 1per mill) during summer and higher during winter-please explain.

Material and methods - please provide a more complete description of the lithology of the record. Are there any discernible flood layers present? If so, does the frequency and/or the flood layer thickness increase during the respective time interval?

[Figure]

- have event layers (e.g. flood layers) been removed prior to the construction of the age model? - a new age-modelling algorithm has been applied to the Lake Petit core- please provide an age-model figure.

Discussion - p. 6 l. 3-6: Why start with human impacts if you can rule those out for the respective time interval? Emphasizing the different factors influencing the hydrological setting is more important in the context of the study- I suggest to start the discussion with those.

- Somewhere in the discussion (and in the site description section) it would be worth noting that the water residence time is short.

- P. 6, L. 27-28: 'Today, Mediterranean precipitation favours runoff and erosion in steep areas (Kosmas et al., 2002)'. Please specify more precisely what type of precipitation favours (intense) runoff and flooding. Also the seasonal distribution of this type of precipitation is important here.

- P. 6, L. 28-31: 'Geochemical data showing high terrigenous inputs to Petit Lake between 4400 and 4000 cal. BP (Fig. 4), interpreted as an increase of runoff in the watershed (Brisset et al., 2013), are thus consistent with a greater seasonal variability of the Mediterranean climate characterised by intense precipitation occurring in fall and spring and significantly drier periods in the summer months (Durand et al., 2009)'.

The statement of changes in seasonality is not supported by the data. Wouldn't an increase in convective precipitation during summer with Mediterranean moisture sourcing also explain both an increase in $\partial$18O and catchment erosion induced by heavy precipitation events. Also, snow cover in early spring would probably inhibit catchment erosion, leaving only heavy precipitation events in summer and early fall to explain an increased erosion pulse.

- P.7, l. 6-8: 'In summary, the rapid increase in $\delta$18O diatom from 4400 to 3900 cal. BP is most likely the result of an increase in water evaporation possibly associated with a

shift in precipitation origin and distribution over the year. This state lasted for ca. 500 years'.

I am not sure I can follow the reasoning here entirely as it is also in part contradictory to the statements made earlier on in the discussion. For example, on page 6 you explain the increase in catchment erosion by an increase in spring and fall precipitation (that is similar compared to today), now here you propose 'a shift in distribution over the year'. Also stronger evaporation is suggested as the main cause for the observed $\partial 18O$ signal. However, the increase in erosion is probably best explained by more frequent and intense summer precipitation events and/or local expansion of glaciers/icefields (glacial cirque just above the lake). I think this is not all wrong but I would suggest the authors to 1) take a look at other records aiming at heavy precipitation reconstructions in nearby sites for the respective time interval, what do the authors of those studies suggest in terms of precipitation type and seasonal distribution? 2) some studies have suggested moderate glacier advances during this period. Wouldn't persistence of snow/ice throughout the summer also influence the hydrological budget of the lake? And at the same time deliver erodible substrates to the lake? The lake is located just below a glacial cirque which appears to have been active not too long ago. I suggest expanding on this somewhat as this is central to the interpretation of the dataset presented.

- P. 7, l. 27-28: Based on the interpretation suggestions above chemical weathering of soils is unlikely to intensify during the proposed climate conditions. Rather soils that formed during wetter and warmer climate phases prior to the 4.2ka event were subject to erosion, resulting in the input of more weathered soil material into Lake Petit. Please revise.

- P. 8, l. 12-29: This paragraph is simply a listing of quotes from references. Please integrate these with your data in a discussion style.

Conclusions
- p. 9, l. 10-11: 'The new $\delta18O$diatom record for Petit Lake was used to reconstruct past hydrological changes and decipher climatic implications from local human impacts around 4200 cal. BP.'

The study focuses on reconstructing hydrological changes, it does not touch upon human impacts. Please revise.

- p.9, l. 23-25: 'This isotopic record at Petit Lake has revealed the implication of the 4.2 kyrs event in abrupt ecosystem changes in the Southern Alps and is useful to better understand the intensity and geographical extent of this climatic event in the Mediterranean region.'

Again, this study, as is, focuses almost exclusively on hydrological changes. If the authors would like to include impacts of hydrological change on ecosystem changes than this part has to be developed throughout the manuscript and not only in the conclusions.

Technical comments:

Not being a native speaker myself I have the impression that this MS can benefit from language polishing by a native speaker. Below please find a few suggestions on how to improve the text.

- p.2, l. 13: In Central Mediterranean, while speleothems from southern Italy (Renella, Corchia Cave) recorded dry conditions from ca. 4300 cal. BP to 3800 cal. BP, dry conditions were less expressed in records from northern Italy.

Revise as follows: In the Central Mediterranean the pattern is less conclusive with dry conditions recorded from ca. 4300 cal. BP to 3800 cal. BP in speleothems from southern Italy (Renella, Corchia Cave)...

- p.2, l. 15: Is this statement true for the entire Alps? Hölloch Speleothems etc and lake sediment records showing the same pattern? Please specify.

- p.2, l. 17-19: 'transition' is used twice in sentence, maybe revise?

- p.3, l. 12: Please specify seasonal rainfall distribution, how much precipitation (in mm) during summer and winter months (fall? Spring?).

- p.3, l. 17: during instead of by, delete ice (meltwater originates from the melting of ice or snow).

- p.3, l. 19: delete being

- p.3, l. 22: in instead of at

- p.4, l. 3: rather: . . .. but rather vary due to changes in source and temperature..

- p.4, l. 5: revise to: . . .after the snow melt..

- p.4, l. 18: Please insert concentrations for H2O2 and HCl.

- p.4, l. 31: ..placed in an..

- p.5, l. 15: Oxygen isotope values not oxygen isotopes values

- p.6, l. 12: frustules instead of spicules

- p.6, l. 15: 'snow' rather than 'ice'

- p.6, l. 16: drops by instead of decreasing

- p.6, l. 20: revise sentence- During summer, waters are well mixed down to the bottom due to wind stress on the open lake surface (Cartier, 2016).

- p.6, l. 32: delete fractionation.

- p.6, l. 33: would require instead of would imply

- p.7, l. 5: delete month

- p.7, l. 7: ..increase in lake water.. .. precipitation sourcing and seasonal distribution..

- p.7, l. 9-10: . . .suggesting more humid conditions..

- p.7, l. 18: rainfall instead of humidity, sourcing instead of influences

- p.8, l. 10-11: Revise sentence, it is rather hard to understand what you want to say. What is a lake trajectory?

- Fig.2: Use one font size and style for axis labels and numbers.

- Fig. 5: Botrychium is not mentioned in text but shown in figure. Either amend text and describe the meaning of it or remove from figure.

---

## Author Comment (AC1) · 18 Dec 2018

*RW1: This paper presents new data from 4.2ka from a part of the world where data from this time are lacking. The data collection (e.g. purification of diatom samples) and the treatment of uncertainty in the age model is thorough and allows us to have more confidence in the results. The introduction presents a good hypothesis based on the previously produced proxies to test in this paper with the d18Odiatom data. However, my main concern is regarding the interpretation of the (slight) increase in d18Odiatom around 4.2ka as primarily a water balance signal (i.e. indicating a shift to drier conditions).*

*But, and the authors acknowledge this at points, a change in precipitation source could also account for some of the isotope shift. In fact, potentially it could account for all of the isotope shift. Also, a decrease in snow (with its very low d18O) around 4.2ka with everything else staying the same could account for the d18Odiatom rise. While your argument about temperature not being the main driver if valid, more thought and caution needs to go in to your interpretation.*

*There are only a couple of modern day lake water isotope values, but even the summer one is fairly low, so how evaporatively-driven is the isotope system? While I agree that something definitely happened in the lake 4.2ka as the d18Odiatom changes are outside of uncertainty and other proxies show changes too, with so few d18Odiatom data points and the relatively small magnitude of the change means it is difficult to unequivocally say that a change to drier conditions, rather than a change in precipitation source, or decrease in snow, or a combination of these factors, was responsible for the d18Odiatom change.*

*Therefore, I think the argument of the driver(s) of d18Odiatom needs to be more cautious and more thought through.*

*Nevertheless, this is a valuable new dataset that is robustly analysed and adds to our knowledge of what was going on in the Mediterranean region around 4.2ka, so I support its publication if my points are addressed.*

**AC: We thank reviewer#1 for these very constructive and meaningful comments concerning the interpretation of the $\delta^{18}O$ record. The factors that may have controlled the $\delta^{18}O_{\text{lake water}}$ and $\delta^{18}O_{\text{diatom}}$ signatures at 4.2 ka cal. BP are reviewed more thoroughly by reorganizing the discussion section as follows:**

1) **A statement on the parameters that may control the present day $\delta^{18}O_{\text{lake water}}$ is made:**

Water inflows to Lake Petit consist of direct precipitation (rain and snow) and intermittent streams that form during the spring snowmelt. There is no groundwater input into the lake and no glacier is present in the watershed, the last period of active glacier advances in the Maritimes Alps being recorded during the Little Ice Age (Ribolini et al., 2007).

The outlet of Lake Petit is an intermittent surface outlet and is non-active when the lake level drops by 1 meter. Therefore, the hydrological regime alternates

between two states: an open system when the outlet is active during snow melt and a closed system during summer months when most water losses are due to evaporation. The 2011 one off $\delta^{18}O_{lake\ water}$ measurements indicate that from the beginning of the unfreezed season to the end, the lake water gets heavier by 1.1 ‰. This $^{18}O$-enrichment may come from the inputs of heavy summer precipitation fed by the Mediterranean Sea (weighted annual mean of -4 ‰ in the Alps compared to -8‰ for precipitation originated from Atlantic) and from evaporation of the lake water. The decrease in water depth during the same time supports a strong evaporation. However, in a $\delta D$ *vs* $\delta^{18}O$ diagram, the lake water samples plot on the regional meteoric water line, which suggests that evaporation has a limited effect on the isotope composition of the lake water. The 1.1‰ shift may also be explained by the drastic decrease of meltwater input at the end of spring. The oxygen isotope composition of meltwater is controlled by $\delta^{18}O$ precipitation, which is lower during winter as the water vapour originates above the Atlantic Ocean (weighted annual mean of -8 ‰ in the Alps), and post-depositional fractionating processes (including evaporation, sublimation, ablation, meltwater percolation and drifting) leading to $^{18}O$ enrichment of the snow. However, because the Lake Petit watershed is small (area of 6 km$^2$) and located under the mountain crest, these post-depositional processes are expected to be of minor importance on the $\delta^{18}O$ of meltwater (Stichler and Schotterer, 2000).

Finally, although only two $\delta^{18}O_{lake\ water}$ measurements are available, they suggest that in the context of current climate conditions, seasonal changes in precipitation sources (i.e. winter Atlantic source vs summer Mediterranean source) leading to significant changes in $\delta^{18}O$ precipitation may control the seasonal shift in $\delta^{18}O_{lake\ water}$.

**2) The $\delta^{18}O_{diatom}$ record around 4.2 ka cal. BP is interpreted in light of i) the modern behavior of the lake, ii) the other climate proxy data from the same core (Cartier et al., 2015) and iii) previous climate reconstructions from the Mediterranean area.**

The 4400 to 3900 cal. BP period is characterized by the highest $\delta^{18}O_{diatom}$ values recorded over the last 4800 years in Lake Petit sediments. These values are about 3 ‰ higher than the modern one (27.8 ‰ in 1986 AD) but correspond to a 1.6 ‰ increasing shift from 4800 to 4400 cal. BP and a 1.5 ‰ decreasing shift from 3900 cal. BP. $\delta^{18}O_{diatom}$ depends on the $\delta^{18}O_{lake}$ and the temperature at which silica polymerizes. The $\delta^{18}O_{lake}$ value is itself influenced by the $\delta^{18}O_{precipitation}$ (rainfall or snow). $\delta^{18}O_{precipitation}$ is controlled by the isotope composition of the vapour source and Rayleigh fractionation during the vapour transport (i.e. the continental and altitude effect) and air temperature at the locality where precipitation forms. Changes in these parameters may combine to account for the high $\delta^{18}O_{diatom}$ values observed from 4400 to 3900 cal. BP at lake Petit. They are reviewed below, in light of the other climate proxy data from the same core (Cartier et al., 2015) and previous climate reconstructions from the Mediterranean area.

*Shift in lake water temperature*

Polymerization of the siliceous frustule from the lake water occurs at equilibrium and the resulting isotope fractionation is thus thermo-dependent. Diatom blooms

in alpine lakes occur mainly after the snowmelt in spring season and during autumn. However, sediment traps placed in a lake in Switzerland located at 2339 m a.s.l have evidence that some diatom species (e.g. *Achnanthes*, *Fragilaria* spp.) can continue to grow under the ice when the lake is frozen (Rautio et al., 2000; Lotter and Bigler, 2000). In the following discussion, we assume that the isotopic signal from Lake Petit sediments is an annual signal even if most of the diatom production most likely occur during the ice-free season.

The equilibrium fractionation coefficient previously measured for different silica-water couples range from -0.2 to -0.4 ‰/°C (synthesis in Alexandre et al., 2012; Sharp et al., 2016). According to this range, a 1.6 ‰ shift in $\delta^{18}O_{diatom}$ only controlled by a lake water temperature change would require a mean annual water temperature shift of 4 to 8°C. Reconstruction of temperature based on chironomids and pollen assemblages from the Swiss Alps and Europe suggest that air temperature variations (likely larger than water temperature variations) did not exceed 2 °C during the Holocene (Davis et al., 2003; Heiri et al., 2003).Thus, although a decrease in mean annual temperature may have contributed to the 4400/3900 cal. BP increase in $\delta^{18}O_{diatom}$, it cannot be the only factor explaining this change. According to studies on speleothems in central Italy (Isola et al., 2018), a cooling during the 4.2 ka BP event in response to a positive North Atlantic Oscillation (NAO) is plausible in central Mediterranean. The recent synthesis of Bini et al. 2018 also suggest the presence of a cooling anomaly but temperature data are sparse and not uniform. In the Alps, moraine dated around 4200 cal. BP showed moderate glacier advances in northern and central western Alps but not in the Maritime Alps (Le Roy, 2012; Ivy-Ochs et al., 2009).

*Shift in $\delta^{18}O_{lake\ water}$*

An increase in the contribution of $^{18}O$-enriched Mediterranean precipitation during the ice-free season, or a $^{18}O$-depleted winter snow deficit may explain an increase in $\delta^{18}O_{lake\ water}$ at Lake Petit at 4400 cal. BP. High terrigenous inputs from 4400 cal. BP support the increase of $^{18}O$-enriched precipitation during the ice-free season. Sedimentological data from the same core (Brisset et al., 2013), allowed to reconstruct before 4400 cal. BP a period of low detrital supply and high chemical weathering from acid soils developed on the slopes. The terrigenous inputs were interpreted as resulting from the dismantling of these weathered soils. The high representation of very low-dispersal alpine meadow pollen (e.g. *Botrychium*) in the sediment additionally argued for an intensification of runoff on the catchment slopes. Similar detrital events were recorded between 4500 and 3000 cal. BP in the Alps, for example at Lake Bourget (Arnaud et al., 2005; 2012). Moreover, a cluster of dated landslide events in the Southern Alps around 4200 cal. BP was interpreted as increasing intense fall precipitation (Zerathe et al., 2014).

A winter $^{18}O$-depleted winter snow deficit can also be suggested. But a oxygen isotope record from speleothems record in Italian Apenin, at Corchia Cave, suggest reduced advection of air masses from the Atlantic during winter from ca. 4.5 to 4.1 ka cal. BP.

An evaporation, higher than the modern one, may also account for a $^{18}O$ enrichment of the surficial water at Lake Petit. However, on an annual basis, the effect of the previous summer's evaporation might be partially or (greatly) offset by the runoff from snowmelt (Ito et al., 2018), as evidenced today.

At least, an increase of air temperature may have led to the increase of the $\delta^{18}O$ of precipitation feeding the lake water. However, as previously discussed, this is not in agreement with other reconstructions from the Mediterranean area that rather argue for a cooling anomaly, although data are scarce (Bini et al. 2018).

**Finally, the shift in $\delta^{18}O_{diatom}$ between 4400 and 3900 cal. BP rather suggests an increase in the contribution of $^{18}O$ enriched Mediterranean precipitation to Lake Petit during the ice-free season. This is in line with increased erosion in the watershed and increased terrigenous inputs to the lake. This does not exclude winter snow deficit and/or summer evaporation and/or on an annual basis, general drier conditions as suggested by Isola et al. (2018). A decrease in annual temperature of the lake water may also have played concomitantly. However, the record from Lake Petit does not allow to further discuss the relative weight of these parameters.**

References
Bini, M., Zanchetta, G., Persoiu, A., Cartier, R., Català, A., Cacho, I., Dean, J. R., Di Rita, F., Drysdale, R. N., Finnè, M., Isola, I., Jalali, B., Lirer, F., Magri, D., Masi, A., Marks, L., Mercuri, A. M., Peyron, O., Sadori, L., Sicre, M.-A., Welc, F., Zielhofer, C., and Brisset, E.: The 4.2 ka BP Event in the Mediterranean Region: an overview, Clim. Past Discuss., in review, 2018.
Isola, I., Zanchetta, G., Drysdale, R. N., Regattieri, E., Bini, M., Bajo, P., Hellstrom, J. C., Baneschi, I., Lionello, P., Woodhead, J., and Greig, A.: The 4.2 ka BP event in the Central Mediterranean: New data from Corchia speleothems (Apuan Alps, central Italy), Clim. Past Discuss., in review, 2018.
Ito, E., Yu, Z., Engstrom, D. R., & Fritz, S. C. (1998). Is paleoclimatic interpretation of oxygen isotope records from glaciated Great Plains possible. Abstracts, AMQUA, 15, 119.
Lotter, A. F., & Bigler, C. (2000). Do diatoms in the Swiss Alps reflect the length of ice-cover? Aquatic sciences, 62(2), 125-141.
Rautio, M., Sorvari, S., & Korhola, A. (2000). Diatom and crustacean zooplankton communities, their seasonal variability and representation in the sediments of subarctic Lake Saanajärvi. Journal of Limnology, 59(1s), 81-96.
Ribolini, A., Chelli, A., Guglielmin, M., & Pappalardo, M. (2007). Relationships between glacier and rock glacier in the Maritime Alps, Schiantala Valley, Italy. Quaternary Research, 68(3), 353-363.
Stichler, W., & Schotterer, U. (2000). From accumulation to discharge: modification of stable isotopes during glacial and post‐glacial processes. Hydrological Processes, 14(8), 1423-1438.

---

## Author Comment (AC2) · 18 Dec 2018

*RW2: Cartier et al. present a novel diatom ∂18O dataset spanning the past ~5000 yrs from Lake Petit in the SW French Alps. The focus of the study lies in the local to regional characterization of hydroclimate perturbations around the 4.2ka climatic event and is thus relevant within the scope of CP. The oxygen isotope data (4 data points for the time slice) show a clear excursion towards higher values, which the authors primarily interpret to be the result of drier conditions with increased evaporation in the Lake Petit watershed. By using already published data from a previous 'multiproxy' study of the Petit sediment record the authors further suggest that the period was characterized by precipitation induced flood events. While the interpretation of the oxygen isotope data appears mostly sound (detailed comments below) it is sometimes hard to follow the argumentation regarding the sedimentary indicators that suggest a higher frequency of flood events/catchment erosion during this period. Since this is quite a central statement for the hydrological reconstructions I would suggest the authors provide a complete lithostratigraphic account of the record (eg. Are there any discernible or identifiable flood layers?). In a broader sense the manuscript contributes an additional hydroclimatic dataset that will help to paint a regional picture of climate repercussions during the 4.2ka event in the Mediterranean borderlands. The manuscript is in most parts appropriately structured, in some parts appropriately illustrated, but suffers from a large number of spelling mistakes and grammatical flaws.*

AC: We thanks the reviewer for its valuable comments to improve the quality of the manuscript. In general, the reviewer highlights the overall quality of the manuscript, the relevance of the dataset presented, and pertinence of the interpretations as a significant new contribution to draw a more precise picture of the 4.2 ka BP event. Also, the reviewer points out several needs for clarifying the manuscript:

- To better describe the previous published dataset on core PET09P2, in order to assist, and restrict, the interpretations of the $d^{18}O$ data;
- To improve the overall writing quality of the manuscript.

We fully agree the reviewer's comments (all details given below), and will modify the manuscript in accordance.

We particularly realized that the term "flood" is somewhat confusing in the manuscript, compared to the recent similar literature in paleolimnology. The term "flood" was mentioned twice in the manuscript (chapter 5.2, lines 6 and 17): first to designate a hydrological process (water overflow of a river channel); second to describe a sedimentological facies (minerogenic normal-graded layer). No "flood layers" are deposited in PET09P2 at the difference of some other lake sediment records of this region, noticeably in Lake Allos that has been the most recently investigated for that purpose (Brisset et al., 2017; Wilhelm et al., 2012). At Lake Allos, the flood layer facies correspond to 60% of the material deposited over the last 7000 yrs. Comparing those two records, Lake Petit and Lake Allos is not straightforward, and probably, has led to the confusions noted by the reviewer.

Those clarifications will be done in the manuscript by including a detailed description of the lithostratigraphy of the Lake Petit and sedimentation processes, referring more precisely to the complementary dataset published by Brisset et al. (2013).

*RW2: Site settings - the seasonal distribution is quite important in this setting. If possible provide precipitation data for summer and winter months. Also, what controls winter snow depth in this setting? From the data presented it seems as if snow depth (by the end of the season?) varies largely from year to year.*

AC: Based on the meteorological station of Malaussène (500 m a.s.l; the closest station to Lake Petit) that covers the period 1997-1998, the precipitation regime in this area is characterized by a marked intra-annual variability, because of the influence of the Mediterranean climate regime. Precipitation essentially occur in spring and autumn (an average of 80 % of the total precipitation volume of the year, corresponding to 758 mm). Snow cover duration is about 185 days at the altitude of Lake Petit from November to April (Durand et al., 2009).
The origin of precipitation vary along the year: while rainfall has a Mediterranean origin (54% of the rainfall events, Celle-Jeanton, 2001), winter snowfalls are essentially associated with northwest atmospheric flows (Durand et al., 2009).

An ombrothermic diagram will be added in the Fig. 1 to make it clear for readers.

*RW2: p.4, l. 1-4. Temperature dependent fractionation of rainfall is suggested as the main driver of seasonal oxygen isotopic composition. However, $\partial 18O$ of precipitation at Malaussene is lower (by almost 1per mill) during summer and higher during winter-please explain.*

AC: Thanks to have point out this mistake. According to the GNIP database, the mean $\delta^{18}O_p$ at the meteorological station of Malaussène is of -4.9 ‰ in summer and of -5.8 ‰ in winter.

*RW2: Material and methods - please provide a more complete description of the lithology of the record. Are there any discernible flood layers present? If so, does the frequency and/or the flood layer thickness increase during the respective time interval?*
*- have event layers (e.g. flood layers) been removed prior to the construction of the age model? - a new age-modelling algorithm has been applied to the Lake Petit core please provide an age-model figure.*

AC: The sediments of the core PET09P2 consist in changes in the relative abundance of a biogenic silica lacustrine production (diatoms), an organic production (essentially algal, e.g. Hydrogen-index comprised between 450 and 575 HC/TOC), and a terrigenous minerogenic clay fraction (Brisset et al., 2012; 2013). The sediments deposited during the period at 4.2 ka are characterized by a higher proportion (80%) and a higher flux of the terrigenous fraction (Brisset

et al., 2013), while the diatom-organic component drop to lower but still significant concentrations (20%).

This unit does not correspond to one or to a cluster of "flood layers", as defined by sedimentological criteria (e.g. Mulder and Chapron, 2011; Gilli et al., 2013): grain-supported sediments, having a distinct – possibly erosional - contact with the previously deposited sediments, and characterized by a normal-graded grain-size sequence. No "flood layers" are deposited in PET09P2 at the difference of some other lake sediment records of this region, noticeably in Lake Allos that is the most recently investigated one on that topic (Brisset et al., 2017; Wilhelm et al., 2012); at Lake Allos, flood layer sedimentological facies, corresponds to 60% of the material deposited over the last 7000 yrs. Characteristics of the lake catchment likely explain the total absence of flood layers. Catchment slopes of Lake Petit are smoothly eroded (morphologies inherited of glacier abrasion processes shaping the glacial step in resistant crystalline rocks), and the gradient of the main river stream is relatively low (10°). These topographic characteristics do not favor surface water concentration to generate a sufficient-water discharge to carry coarse particle.

To clarify those points, the detailed lithological information will be added in the manuscript and the Fig. 4 will be complemented of the lithostratigraphic log.

The age-depth model presented in this study is indeed a new algorithm not published yet, and part of the present paper. The value of the algorithm "bacon" developed by Blaauw and Christen (2011) is to calculate the age probability density function of the data proxy. In the present paper, applying this approach is necessary to demonstrate that the 4.2 event is well constrained in the PET09P2 core (that is a minimum), and interestingly plus: the event cannot be instantaneous in time, and its time range is at a confidence interval of 95% of probability of > 25 years and < 660 years. Given these valuable results, we decided to recalculate the model (done using the "clam" R package in Brisset et al., 2013) by a model calculated using the 'bacon' R package (Blaauw and Christen, 2011).

We agree that adding the all details of this new model in the manuscript will contribute to a better understanding of the overall dataset and include those information in the revised manuscript.

*RW2: Discussion - p. 6 l. 3-6: Why start with human impacts if you can rule those out for the respective time interval? Emphasizing the different factors influencing the hydrological setting is more important in the context of the study- I suggest to start the discussion with those.*

AC: Following the comment of the reviewer, the first sentences concerning human impacts have been removed. The discussion starts with a description of the rise in $\delta^{18}O_{diatom}$ during the 4.2 ka BP event and the main factors influencing oxygen isotopes (changes in water temperature and $\delta^{18}O_{lake\ water}$).

*RW2: Somewhere in the discussion (and in the site description section) it would be worth noting that the water residence time is short.*

AC: We will add in the site settings that according to the size of Lake Petit the water residence time is expected to be short even if we don't have a quantitative estimate.

*RW2: P. 6, L. 27-28: 'Today, Mediterranean precipitation favours runoff and erosion in steep areas (Kosmas et al., 2002)'. Please specify more precisely what type of precipitation favours (intense) runoff and flooding. Also the seasonal distribution of this type of precipitation is important here.*

AC: According to a synthesis of floods and flash flood events in Mediterranean countries different types of precipitation favour intense runoff: short and local summer flash flood event, autumn high-rainfall event and extended rainfall event affecting more than one country (Llasat et al., 2010). In this study 185 flood events (daily accumulated precipitation over 60 mm) on the period 1990-2006 were distributed as follows: 54.7 % of the annual total occurred in autumn (September, October and November) while the summer months have 17.2 % and winter 15.3 %. In addition, Descroix et al. (2010) has shown that soil erosion is higher after long periods of drought.
These more detailed information will be inserted in the manuscript.

*RW2: P. 6, L. 28-31: 'Geochemical data showing high terrigenous inputs to Petit Lake between 4400 and 4000 cal. BP (Fig. 4), interpreted as an increase of runoff in the watershed (Brisset et al., 2013), are thus consistent with a greater seasonal variability of the Mediterranean climate characterised by intense precipitation occurring in fall and spring and significantly drier periods in the summer months (Durand et al., 2009)'.*
*The statement of changes in seasonality is not supported by the data. Wouldn't an increase in convective precipitation during summer with Mediterranean moisture sourcing also explain both an increase in @18O and catchment erosion induced by heavy precipitation events. Also, snow cover in early spring would probably inhibit catchment erosion, leaving only heavy precipitation events in summer and early fall to explain an increased erosion pulse.*

AC: Higher terrigenous inputs to Lake Petit during the 4.2 ka BP event highlighted the presence of intense precipitation events during the ice-free season which last in average from April to October. Therefore, we support the hypothesis that the precipitation regime has changed during this period of the year to produce the changes in erosion processes. According to Durand et al. (2009), precipitation in southern Alps occur mainly in Spring and Autumn. At the scale of the Mediterranean region, 54.7 % of heavy precipitation occur in Autumn (Llasat et al., 2010). Convective precipitation events from Mediterranean moisture sourcing are of higher occurrence during these months when air masses from the Mediterranean, still warm and humid meet the cold air masses from the Atlantic.

We agree with the reviewer that this sentence is mixing several ideas and that changes in seasonality can't be assessed precisely during the ice-free season. For this purpose, we have improved the discussion on the isotopic interpretation by adding sub-sections for each factor of interest including changing in snow

contribution, precipitation regime and sourcing (refer to the answer to the reviewer 1).

*RW2: P.7, l. 6-8: 'In summary, the rapid increase in $\delta^{18}O$ diatom from 4400 to 3900 cal. BP is most likely the result of an increase in water evaporation possibly associated with a shift in precipitation origin and distribution over the year. This state lasted for ca. 500 years'. I am not sure I can follow the reasoning here entirely as it is also in part contradictory to the statements made earlier on in the discussion. For example, on page 6 you explain the increase in catchment erosion by an increase in spring and fall precipitation (that is similar compared to today), now here you propose 'a shift in distribution over the year'. Also stronger evaporation is suggested as the main cause for the observed $\delta^{18}O$ signal. However, the increase in erosion is probably best explained by more frequent and intense summer precipitation events and/or local expansion of glaciers/icefields (glacial cirque just above the lake). I think this is not all wrong but I would suggest the authors to 1) take a look at other records aiming at heavy precipitation reconstructions in nearby sites for the respective time interval, what do the authors of those studies suggest in terms of precipitation type and seasonal distribution? 2) some studies have suggested moderate glacier advances during this period. Wouldn't persistence of snow/ice throughout the summer also influence the hydrological budget of the lake? And at the same time deliver erodible substrates to the lake? The lake is located just below a glacial cirque which appears to have been active not too long ago. I suggest expanding on this somewhat as this is central to the interpretation of the dataset presented.*

AC: According to the geochemical data, the signature of the alumino-silicate fraction indicates high cation fractionation characteristic of pedogenetic origin. At Lake Allos (close to Lake Petit), no evidence of increasing flood frequency has been recorded around 4.2 ka BP but it has been shown that erosion was inhibited prior to deforestation and dismantling of soils by human activities at ca. 1700 cal. BP (Brisset et al., 2017). A synthesis of flood frequency across the Central Alps has shown evidence of increasing flood frequency from 4.2 ka BP to 2.4 ka BP and during the Little Ice Age certainly linked to a southerly position of the N-Atlantic circulation (Wirth et al., 2013). In their study sites, they interpreted flood records to be mainly a record of spring and fall events. These general wetter conditions across the Alps might have been a factor of decreasing $\delta^{18}O_{diatom}$ after the 4.2 ka BP event and during the Little Ice Age (fig. 3). During the 4.2 ka event at Lake Petit, we argue that the intensity of precipitation increased during the ice-free season but not necessarily the occurrence of events. $\delta^{18}O_{diatom}$ values suggest a higher contribution of $^{18}O$ enriched precipitation of Mediterranean origin to the lake water balance (refer to the answer to reviewer 1).

We agree with the reviewer that some studies have recorded moderate glacier advances in central western Alps and in the northern Alps but not, for now, in the Mediterranean Alps (Le Roy, 2012; Ivy-Ochs et al., 2009). According to the last review for the Mediterranean Alps (Brisset et al., 2015) the Holocene re-activation of glaciers has been dated 2720-2360 cal. BP (Ribolini et al., 2007). Rock glacier activities are also recorded later during the Little Ice Age which can explain why the glacier cirque appears to have been active not too long ago

(Federici and Stefanini, 2001). By looking at the isotopic record, the lowest value of $\delta^{18}O_{diatom}$ is during the Little Ice Age possibly due to wetter conditions during this period associated with higher snow contribution from the Atlantic. A persistence of snow during summer which has a low $\delta^{18}O$ signature would most likely lowered $\delta^{18}O_{diatom}$ contrary to what is observed during the 4.2 ka BP event. We will add more reference in the manuscript to reconstructions of glacier advances and heavy precipitation in the Southern Alps.

*RW2: P. 7, l. 27-28: Based on the interpretation suggestions above chemical weathering of soils is unlikely to intensify during the proposed climate conditions. Rather soils that formed during wetter and warmer climate phases prior to the 4.2ka event were subject to erosion, resulting in the input of more weathered soil material into Lake Petit. Please revise.*

AC: In the paper of Brisset et al., 2013 sedimentological data have been interpreted as follows: from 4700 to 4200 cal. BP a period of low detrital supply, high fractionation of cations suggesting the presence of developed acid soils in the watershed and high chemical weathering; during the 4.2 ka BP event: a maximum of clay detrital supply and high fractionation of alumino-silicates highlighting a dismantling of the former developed weathered soils.
We will revise this sentence to say that the high fraction of alumino-silicates during the 4.2 ka BP event is more likely the result of dismantling weathered soils formed during the previous period. A decrease of chemical weathering during the 4.2 ka BP event is possible due to drier conditions.

*RW2: P. 8, l. 12-29: This paragraph is simply a listing of quotes from references. Please integrate these with your data in a discussion style.*

AC: We will improve this part of the manuscript to follow the suggestion of the reviewer.

*RW2: p. 9, l. 10-11: 'The new 18Odiatom record for Petit Lake was used to reconstruct past hydrological changes and decipher climatic implications from local human impacts around 4200 cal. BP.'*
*The study focuses on reconstructing hydrological changes, it does not touch upon human impacts. Please revise.*

AC: This publication is following several papers on Lake Petit (Brisset et al., 2012; Brisset et al., 2013; Cartier et al.,2015) we wanted to highlight the fact that the use of oxygen isotopes from diatoms allowed us to confirm the effect of climate on the environmental change observed at Lake Petit at that time.

*RW2: p.9, l. 23-25: 'This isotopic record at Petit Lake has revealed the implication of the 4.2 kyrs event in abrupt ecosystem changes in the Southern Alps and is useful to better understand the intensity and geographical extent of this climatic event in the Mediterranean region.'*
*Again, this study, as is, focuses almost exclusively on hydrological changes. If the authors would like to include impacts of hydrological change on ecosystem changes*

*than this part has to be developed throughout the manuscript and not only in the conclusions.*

AC: We will change this sentence by "This isotopic record confirms the implication of the 4.2 ka BP event in the environmental responses observed at Lake Petit in previous studies (Brisset et al., 2013; Cartier et al., 2015)". The summary of environmental changes during the 4.2 ka event is present in part 5.2. The detailed results of environmental responses are presented in previous papers.

References

Brisset, E., Guiter, F., Miramont, C., Delhon, C., Arnaud, F., Disnar, J.-R., Poulenard, J., Anthony, E., Meunier, J.-D., Wilhelm, B., Pailles, C., 2012. Approche multidisciplinaire d'une séquence lacustre holocène dans les alpes du sud au Lac Petit (Mercantour, alt. 2 200 m, France) : histoire d'un géosystème dégradé. Quaternaire. Revue de l'Association française pour l'étude du Quaternaire 23, 309–319.

Brisset, E., Miramont, C., Guiter, F., Anthony, E.J., Tachikawa, K., Poulenard, J., Arnaud, F., Delhon, C., Meunier, J.-D., Bard, E., Suméra, F., 2013. Non-reversible geosystem destabilisation at 4200 cal. BP: Sedimentological, geochemical and botanical markers of soil erosion recorded in a Mediterranean alpine lake. The Holocene 23, 1863–1874.

Brisset, E., Guiter, F., Miramont, C., Troussier, T., Sabatier, P., Poher, Y., Cartier, R., Arnaud, F., Malet, E., Anthony, E.J., 2017. The overlooked human influence in historic and prehistoric floods in the European Alps. Geology 45, 347–350.

Descroix, L. (2003). Les conséquences hydrologiques de l'évolution des usages des sols, 78 pp. Université Joseph Fourier, Grenoble Mémoire d'Habilitation à Diriger des Recherches (HDR).

Federici, P. R., & Stefanini, M. C. (2001). ABHANDLUNGEN-Evidence and chronology of the Little Ice Age in the Argentera Massif (italian maritime alps). With 7 figures. *Zeitschrift fur Gletscherkunde und Glazialgeologie*, *37*(1), 35-48.

Gilli, A., Anselmetti, F.S., Glur, L., Wirth, S.B., 2013. Lake Sediments as Archives of Recurrence Rates and Intensities of Past Flood Events, in: Schneuwly-Bollschweiler.

Ivy-Ochs, S., Kerschner, H., Maisch, M., Christl, M., Kubik, P. W., & Schlüchter, C. (2009). Latest Pleistocene and Holocene glacier variations in the European Alps. *Quaternary Science Reviews*, *28*(21-22), 2137-2149.

Le Roy, Melaine. *Reconstitution des fluctuations glaciaires holocènes dans les Alpes occidentales: apports de la dendrochronologie et de la datation par isotopes cosmogéniques produits in situ*. Diss. Grenoble, 2012.

M., Stoffel, M., Rudolf-Miklau, F. (Eds.), Dating Torrential Processes on Fans and Cones, Advances in Global Change Research. Springer Netherlands, pp. 225–242.

Llasat, M. C., Llasat-Botija, M., Prat, M. A., Porcu, F., Price, C., Mugnai, A., ... & Yair, Y. (2010). High-impact floods and flash floods in Mediterranean countries: the FLASH preliminary database. *Advances in Geosciences*, *23*, 47-55.

Mulder, T., Chapron, E., 2011. Flood deposits in continental and marine environments: Character and significance, Sediment Transfer from Shelf to Deep Water: Revisiting the Delivery System, AAPG Studies in Geology. Slatt M. et Zavala C.

Ribolini, A., Chelli, A., Guglielmin, M., & Pappalardo, M. (2007). Relationships between glacier and rock glacier in the Maritime Alps, Schiantala Valley, Italy. *Quaternary Research*, *68*(3), 353-363.

Wilhelm, B., Arnaud, F., Sabatier, P., Crouzet, C., Brisset, E., Chaumillon, E., Disnar, J.-R., Guiter, F., Malet, E., Reyss, J.-L., Tachikawa, K., Bard, E., Delannoy, J.-J., 2012. 1400 years of extreme precipitation patterns over the Mediterranean French Alps and possible forcing mechanisms. Quaternary Research 78, 1–12.

Wirth, S. B., Glur, L., Gilli, A., & Anselmetti, F. S. (2013). Holocene flood frequency across the Central Alps–solar forcing and evidence for variations in North Atlantic atmospheric circulation. *Quaternary Science Reviews*, *80*, 112-128.